# Genes Involved by Dexamethasone in Prevention of Long-Term Memory Impairment Caused by Lipopolysaccharide-Induced Neuroinflammation

**DOI:** 10.3390/biomedicines11102595

**Published:** 2023-09-22

**Authors:** Galina T. Shishkina, Tatyana S. Kalinina, Dmitriy A. Lanshakov, Veta V. Bulygina, Natalya P. Komysheva, Anita V. Bannova, Ulyana S. Drozd, Nikolay N. Dygalo

**Affiliations:** Laboratory of Functional Neurogenomics, Federal Research Center Institute of Cytology and Genetics, Siberian Branch of the Russian Academy of Science, Novosibirsk 630090, Russia; kalin@bionet.nsc.ru (T.S.K.); lanshakov@bionet.nsc.ru (D.A.L.); veta@bionet.nsc.ru (V.V.B.); agarina@bionet.nsc.ru (N.P.K.); anitik@bionet.nsc.ru (A.V.B.); drozd@bionet.nsc.ru (U.S.D.); dygalo@bionet.nsc.ru (N.N.D.)

**Keywords:** lipopolysaccharide, dexamethasone, hippocampus, RNA-sequencing, memory, inflammation, glutamate

## Abstract

Inflammatory activation within the brain is linked to a decrease in cognitive abilities; however, the molecular mechanisms implicated in the development of inflammatory-related cognitive dysfunction and its prevention are poorly understood. This study compared the responses of hippocampal transcriptomes 3 months after the striatal infusion of lipopolysaccharide (LPS; 30 µg), resulting in memory loss, or with dexamethasone (DEX; 5 mg/kg intraperitoneal) pretreatment, which abolished the long-term LPS-induced memory impairment. After LPS treatment, a significant elevation in the expression of immunity/inflammatory-linked genes, including chemokines (*Cxcl13*), cytokines (*Il1b* and *Tnfsf13b*), and major histocompatibility complex (MHC) class II members (*Cd74*, *RT1-Ba*, *RT1-Bb*, *RT1-Da*, and *RT1-Db1*) was observed. DEX pretreatment did not change the expression of these genes, but significantly affected the expression of genes encoding ion channels, primarily calcium and potassium channels, regulators of glutamate (*Slc1a2*, *Grm5*, *Grin2a*), and GABA (*Gabrr2*, *Gabrb2*) neurotransmission, which enriched in such GO biological processes as “Regulation of transmembrane transport”, “Cognition”, “Learning”, “Neurogenesis”, and “Nervous system development”. Taken together, these data suggest that (1) pretreatment with DEX did not markedly affect LPS-induced prolonged inflammatory response; (2) DEX pretreatment can affect processes associated with glutamatergic signaling and nervous system development, possibly involved in the recovery of memory impairment induced by LPS.

## 1. Introduction

Numerous reports indicate the involvement of inflammatory responses within the brain in the pathogenesis of cognitive impairment in neurodegenerative pathologies, including Alzheimer’s and Parkinson’s diseases [1,2,3,4]. Neuroinflammation was also linked to a decrease in cognitive abilities after traumatic brain injuries [5] and ischemia [6,7,8]. However, the molecular mechanisms underlying the cognitive impairment induced by neuroinflammatory activation, as well as possible ways to prevent or reduce the development of this dysfunction, are poorly understood.

Animal studies showed that the inflammatory response induced by the central administration of lipopolysaccharide (LPS), a cell wall component of Gram-negative bacteria, was accompanied by cognitive damage, as evidenced by an inability to increase the recognition index during the object recognition test in both short (48 h) [9] and delayed (3 months) [10] periods after LPS. The glucocorticoid dexamethasone (DEX) can suppress the immune system and reduce inflammation [11]; therefore, the action of this steroid should reduce neuroinflammation-induced cognitive impairment. Indeed, DEX intraperitoneally administered 30 min before ischemia has been demonstrated to attenuate acute impairment in the cognitive ability of mice and reduce the expression of TLR4, NF-kB, and CD68 in the hippocampus [12].

The identification of changes in brain gene expression after LPS is essential for understanding the molecular pathways involved in the development of inflammation-induced cognitive and affective disturbances [13]. As shown in this and our previous study [14], central injections of LPS resulted in hippocampal gene expression changes at 24 h after endotoxin administration. Considering the important role of the hippocampus in recognition memory, in the present study, we tested the hypothesis that long-term cognitive impairment occurring along with LPS-induced changes in gene expression in the hippocampus can be prevented by DEX. Transcriptomic changes in the hippocampus associated with the long-term behavioral effects of DEX administered before LPS have not yet been reported.

## 2. Materials and Methods

### 2.1. Animals

Sevel-week-old male Wistar rats (weight 150–180 g) at the beginning of the experiments were used in our study. The rats were housed under a 14 h light/10 h dark cycles with free access to food and water. All animal procedures were approved by the ethics committee of the Institute of Cytology and Genetics in accordance with the guidelines of the Russian Ministry of Health regulations on Good Laboratory Practice (supplement to order No. 199n of 1 April 2016) and the European Council Directive (86/609/EEC).

### 2.2. Experimental Protocols

In order to activate neuroinflammation, lipopolysaccharide (LPS) from Escherichia coli serotype 055:B5 (Sigma-Aldrich Corp., St Louis, MO, USA) at 30 µg in 4 µL of sterile saline was infused stereotactically into the right striatum under isoflurane anesthesia (4% isoflurane for induction, 2.5% for maintenance in O_2_ at a flow rate of 1 L/min) using the following coordinates: AP = +0.5 mm, ML = + 3 mm, DV = −5.5/4.5 mm [14,15]. Control animals received an equivalent volume of saline (SAL). Intra-brain LPS exposure can superimpose the damaging effect of the central injection per se [16]. In our work, the region of the apoptosis activation evaluated by tissue volume containing active caspase-3-positive cells determined by serial sections of the right striatum was approximately 0.5 mm^3^ after SAL and 7.3 mm^3^ after LPS showing a very weak impact of the central injection per se.

In the first set of investigations, twenty-four hours after the central administration of LPS (*n* = 6–8) or SAL (*n* = 6–8), rats were quickly decapitated and brain tissue samples (left and right striatum and hippocampus) were collected for mRNAs and proteins of pro-inflammatory and pro-apoptotic markers. Rats (*n* = 4 per group) were used with the same purposes for immunohistochemical analysis.

In the second set of investigations, the rats were randomly divided into two groups (*n* = 20 per group) according to intra-striatal treatment with LPS or SAL. Thirty minutes before the central administration of the drugs, half of the animals from each group received intraperitoneal injections of dexamethasone (DEX; KRKA Slovenia; 5 mg/kg) or vehicle (dimethyl sulfoxide; DMSO; Sigma Aldrich USA). Four groups of animals were created: LPS, SAL, DEX + LPS, and DEX + SAL. Twenty-four hours after the drug administration, rats were subjected to the neurological test by Garcia [17]. Three months later, the same rats were tested using the Novel Object Recognition (NOR) test. On the next day after the NOR test, the rats were sacrificed by rapid decapitation. Brains were quickly extracted, and the ipsilateral hippocampi (*n* = 4 for each group) were rapidly isolated and each was placed in an Eppendorf tube with 1 mL of buffer containing an RNase inhibitor (RNAlater) at room temperature. After that, the tube was transferred to ice, and after the end of the hippocampal collection, stored overnight at +4 °C and then at −80 °C until the analysis of gene expression patterns.

### 2.3. Neurological Test

Neurologic testing was performed using the Garcia Neurologic Test [17]. This test is based on the 18-point behavioral scale and includes the evaluation of spontaneous activity, symmetry in the movement of four Limbs, forepaw outstretching, climbing, body proprioception, and response to vibrissae touch (each scored between 0 and 3; the maximum was 18).

### 2.4. Novel Object Recognition (NOR) Test

The NOR test was performed 3 months after drug administration to determine if LPS impacts recognition memory, and if dexamethasone can alter LPS-induced behavioral effects. The test consisted of 3 sessions. During the first session, habituation, the rats were allowed to explore the open field arena (100 cm × 100 cm) for 10 min. In the next session 24 h later, two identical objects were placed on the OF (60 cm apart from each other), and the rats were allowed to explore them for 5 min. After that, rats were kept in the home cage for 1 h, and then a discrimination session was performed, during which the rats were allowed to explore the old and a new object for 5 min. The OF arena was carefully cleaned with ethanol 10% between all testing periods. The time spent by the rats exploring each object and the number of contacts were measured, and in order to analyze the cognitive performance, a recognition index was calculated in each session: the number of contacts with novel object/the number of contacts with both objects.

### 2.5. RNA-Sequencing and Data Analysis

RNA-seq was performed using JSC Genoanalytica (Moscow, Russia; http://genoanalytica.ru). For this, 3 months after drug administration, total RNA was extracted from ipsilateral rat hippocampus with Trisol reagent according to the manufacturer’s instruction (*n* = 4 in each group). Quality was checked using BioAnalyser and RNA 6000 Nano Kit (Agilent, Santa Clara, CA, USA). PolyA RNA was purified using Dynabeads^®^ mRNA Purification Kit (Ambion, Thermo Fisher Scientific, Waltham, MA, USA). Illumina library was made from polyA NEBNext^®^ Ultra™II RNA Library Prep (NEB, Ipswich, MA, USA), according to the manual. Sequencing was performed on HiSeq1500 with a 50 bp read length. At least 20 million reads were generated for each sample.

The raw reads from RNA-seq experiments were trimmed for quality (phred ≥ 20) and length (bp ≥ 32) using Trimmomatic v3.2.2 [18]. Reads were mapped to the Rnor_6.0 genome with STAR aligner [19] and differentially expressed transcripts were inferred by DESeq2.0 [20]. Genes with │log2 FC (fold change)│≥ 1 and an adjusted *p*-value (padj) less than 0.05 were classified as significantly differentially expressed genes (DEGs).

The Database for Annotation, Visualization and Integrated Discovery (DAVID; version 6.8; https://david.ncifcrf.gov/, accessed on 1 December 2022 [21] was used to perform functional annotation.

### 2.6. Real-Time PCR (RT-PCR)

Total RNA was isolated from the brain tissue samples using a one-step guanidine isothiocyanate method. cDNA was synthesized using 3 µg of total RNA via M-MuLV reverse transcriptase (SybEnzyme, Novosibirsk, Russia) and oligo(dT)_15_ for 1 h at 42 °C. The IL-1β mRNA was quantified relative to the actb mRNA using TaqMan^®^ Gene Expression Assay primers/probes (Rn00580432_m1 for IL-1β, and Rn00667869_m1 for actb; Thermo Fisher Scientific, USA) and VIIA™ 7 Real-Time PCR System (Thermo Fisher Scientific, USA). The mRNA levels were calculated using the ΔΔCt method in triplicate for each sample.

### 2.7. Immunofluorescence Assay

One day after LPS or SAL administration into the striatum, four male rats from each group were selected for immunohistochemistry assay. Animals were deeply anesthetized using avertin, and their brains were removed after transcardial perfusion with ice-cold 0.02 M PBS followed by 4% paraformaldehyde (PFA) in 0.02 M phosphate-buffered saline (PBS). The perfused brains were post-fixed in the same fixative at 4 °C for 4 h and then cryoprotected via complete saturation in a 25% sucrose solution in 0.1 M phosphate buffer (PB; pH 7.4) at 4 °C. After that, the brains were quickly frozen using powdered dry ice, cut into 20-μm-thick coronal sections on a cryostat, and stored at −60 °C before immunohistochemistry. The sections were rinsed twice in 0.02 M PBS using 0.1% Triton X-100 (PBST), and nonspecific binding sites were blocked using a 2.5% BSA (bovine serum albumin) in 0.02 M PBST for 1 h at room temperature. Subsequently, sections were incubated with primary antibodies against cleaved Caspase-3 (AB 9664, Cell Signalling, rabbit; dilution 1:200) and Iba-1 (ab 5076, Abcam, goat; dilution 1:200) for 24 h at 4 °C. After that, the sections were washed with 0.02M PBS and incubated with secondary antibodies, F(ab′)_2_ donkey anti-rabbit IgG conjugated with Cy3 (711-166-152, Jackson ImmunoResearch; dilution 1:300) and F(ab′)_2_ donkey anti-goat IgG conjugated with Alexa Fluor 488 (705-546-147, Jackson ImmunoResearch; dilution 1:300) for 2 h at room temperature. For the negative controls, sections were incubated without primary antibodies. All sections were then washed with 0.02 M PBS and mounted using MOWIOL mounting medium with DAPI. Images were acquired on a confocal microscope (LSM 780 NLO) equipped with 405 nm, 488 nm, and 561 nm lasers using a Plan-Apochromat 20 objective (0.8 numerical aperture). Caspase-3 and Iba-1 immunoreactivities were quantified by counting the amount of their immunopositive cells on sections. At least seven sections were analyzed for each rat with the open sourced program QuPath-0.2.3. The striatum regions for the protein expression analyses were determined using the atlas of Paxinos and Watson (from Bregma −0.8 to +1.6 mm).

### 2.8. Western Blot

Western blot was performed as described previously [22]. The brain samples were homogenized in lysis buffer containing 150 mM NaCl, 50 mM Tris, 1% Triton X-100, and protease inhibitors: 2 mM PMSF and 2 μg/mL leupeptin, pepstatin and aprotinin. Protein samples (50 μg) were separated via SDS electrophoresis in Mini-Protean 3 Dodeca Cell (Bio-Rad Laboratories, USA) in 12% polyacrylamide gel. The resolved proteins were transferred to 0.45-µm nitrocellulose membrane by Trans-Blot system (Bio-Rad Laboratories, USA). The proteins were stained with primary antibodies: rabbit monoclonal antibodies Iba-1 (1:500, EPR16589, ab178847, Abcam, Cambridge, MA), cleaved caspase-3 (1:500, #9664, Cell Signaling, USA), and rabbit polyclonal antibodies β-actin (1:20000, I-19, sc-1616, Santa Cruz Biotechnology, USA). Secondary anti-rabbit IgG (Bio-Rad Laboratories, USA) were used at 1:1000 dilutions for Iba-1, cleaved caspase-3 staining, and 1:10000 dilution for β-actin staining. The blots were developed with a SuperSignalTM West Femto Maximum Sensitivity Substrate chemiluminescence kit (Thermo Fisher Scientific, USA) and quantified after scanning with a ChemidocTM Touch Imaging System (Bio-Rad Laboratories, USA) using the Scion Image 4.0.3.2 program (Scion Corporation, USA). The levels of Iba-1 and cleaved caspase-3 proteins were expressed in relative units to β-actin in the same sample.

### 2.9. Statistical Analysis

For Western blot, PCR, immunohistochemical and behavioral data, two-way analysis of variance (ANOVA), and Tukey’s post hoc test were used to analyze between-group differences. Significant differences were set at *p* < 0.05.

## 3. Results

### 3.1. Ecute Effects of LPS on Striatum and Hippocampus

To verify the acute activation of neuroinflammation and neurodegeneration, at 24 h after LPS administration into the right striatum, the expression of Iba1, interleukin-1beta, and cleaved caspase-3 was investigated in the right and left striatum, the ipsilateral, and contralateral hippocampus using real-time PCR, immunoblotting, and immunohistochemically.

#### 3.1.1. Striatum

As shown in Figure 1, LPS caused rapid pro-inflammatory (a) and pro-apoptotic (b) responses in the right striatum, which was a site of endotoxin administration. On the next day after LPS, in the right striatum, PCR and immunoblot analyses showed a significant increase in gene expression of the main pro-inflammatory cytokine interleukin-1beta (effect of LPS: F(1, 22) = 13.897, *p* = 0.0011) (a) and protein of cleaved caspase-3 (effect of LPS: F(1, 28) =3 0.702, *p* = 0.000006) (b) when compared with appropriate SAL groups. At this time point, the effects were not observed in the left striatum (interaction LPS × striatum side: F(1, 22) = 12.379, *p* = 0.0019 for interleukin-1beta; and F(1, 28) = 7.865, *p* = 0.0091 for caspase-3). The full original images of western blots with all legends are presented in the Appendix A.

It is well-known that LPS activates microglial cells. To evaluate the activation of microglia in the right striatum and caspase-3 in these cells after LPS, antibodies to Iba1 (macrophagic-specific calcium-binding protein) and caspase-3 were employed for immunohistochemical staining. Contrary to our expectation, no significant differences were found between the LPS and SAL groups in the number of cells that were immuno-positive only for Iba1. At the same time, the number of cells that showed immunoreactivities for both Iba-1 and caspase-3 was significantly increased after LPS (Figure 1c,d). In contrast to the right striatum, no cells expressing active caspase-3 were found in the left striatum (Appendix A).

#### 3.1.2. Hippocampus

In contrast to the striatum, in which changes in parameters investigated were observed mainly on the side of LPS administration, short-term pro-inflammatory responses to endotoxin were similar in both the right and the left hippocampus. Compared to appropriate controls, PCR and immunoblot analyses showed increased levels of the pro-inflammatory cytokine interleukin-1beta mRNA one day after LPS injection: effect of LPS: F(1, 32) = 21.385, *p* = 0.00006; interaction LPS x hippocampal side: (F(1, 32) = 0.234, *p* = 0.6321) (Figure 2a). However, at this time point, an increase in expression of Iba-1 protein (effect of LPS: F(1, 28) = 8.584, *p* = 0.0067; interaction LPS x hippocampal side: (F(1, 28) = 1.354, *p* = 0.2544) reached statistical significance only in the contralateral hippocampus (Figure 2b).

### 3.2. Behavioral Effects of Intra-Striatal LPS and Pretreatment with DEX at 24 h and 3 Months

#### 3.2.1. A Single Pretreatment with DEX Attenuated the Acute LPS-Induced Neurological Deficit

The Garcia score in animals of the LPS group was significantly lower than that in the SAL group, while the pretreatment of LPS animals with DEX, a classic anti-inflammatory drug, attenuated this decrease. As a result, the neurological score in the DEX + LPS group was higher than in the LPS group (Figure 3a).

#### 3.2.2. A Single Pretreatment with DEX Prevented LPS-Induced Long-Lasting Memory Impairment

In the discrimination session of the NOR test that was performed 3 months after drug administration on the same animals used for neurological deficit evaluation described above, the recognition index was significantly decreased in the LPS group, whereas this effect was absent when DEX was injected 30 min before LPS (interaction LPS × DEX: F(1, 36) = 6.7903, *p* = 0.0132) (Figure 3b).

### 3.3. Transcriptomic Analyses

In an attempt to identify the molecular mechanisms that underlie the delayed behavioral effects of LPS and their modification by DEX pretreatment, transcriptomic analyses of the hippocampus were performed 3 months after the drug administration. To examine the effect of LPS, LPS-administered rats were compared with the SAL group. The effect of DEX pretreatment was examined by comparing DEX + LPS and DEX + SAL, as well as DEX + LPS and LPS groups.

According to the criteria for screening of differentially expressed genes (DEGs): │log2 FC (fold change)│ ≥ 1 and an adjusted *p* value (padj) < 0.05, 33 DEGs, 29 upregulated, and 4 downregulated (*Alox5*, *LOC500035*, *Tbx19*, *H1f4*), were revealed in LPS vs. SAL (Appendix A). The pretreatment of LPS rats with DEX increased the total number of DEGs that were 161 (128 upregulated and 33 downregulated) in DEX + LPS vs. DEX + SAL (Appendix A), whereas no DEGs were detected in DEX+ SAL vs. SAL. The comparison of DEX + LPS and LPS groups revealed 112 DEGs (110 upregulated and 2 downregulated). All DEGs of the three compared pairs of groups, LPS and SAL, DEX + LPS and DEX + SAL, and DEX + LPS and LPS, at the significant level padj < 0.05, but │log2 FC (fold change)│ ≥ 0, presented in volcano plots (Figure 4). According to these criteria, only 3 DEGs, *Ccdc88a*, *Ptprb*, and *RT1-N2* were found in DEX+ SAL vs. SAL (volcano plot not shown).

Using the Venn diagrams (http://bioin forma tics.psb.ugent.be/webtools/Venn/, accessed on 1 April 2023), 22 common genes were revealed between DEGs among 33 DEGs in LPS vs. SAL and 161 DEGs in DEX + LPS vs. DEX + SAL, as well as 84 common genes among 161 DEGs in DEX + LPS vs. DEX + SAL and 112 DEGs in DEX + LPS vs. LPS (Figure 5a; Appendix A). Figure 5b shows the expression profiles of 33 DEGs for LPS vs. SAL. Data were expressed as log2 transformation of the fold changes. Asterisks indicate 22 DEGs shared between LPS vs. SAL and DEX + LPS vs. DEX + SAL.

Gene ontology (GO) analysis showed that DEGs after LPS administration (LPS vs. SAL) were predominantly associated with immune and inflammatory responses (Appendix A; *p*-value < 0.05). The 13 most enriched GO terms for biological processes (FDR (false discovery rate) < 0.05) were “antigen processing and presentation of exogenous peptide antigen via MHC class II” (GO:0019886), “antigen processing and presentation of peptide or polysaccharide antigen via MHC class II” (GO:0002504), “response to interferon-gamma“ (GO:0034341), “peptide antigen assembly with MHC class II protein complex” (GO:0002503), “positive regulation of immune response” (GO:0050778), “positive regulation of T cell activation” (GO:0050870), “adaptive immune response” (GO:0002250), “inflammatory response” (GO:0006954), “antigen processing and presentation” (GO:0019882), “immune response” (GO:0006955), “positive regulation of neutrophil chemotaxis” (GO:0090023), “immunoglobulin mediated immune response” (GO:0016064), and “humoral immune response” (GO:0006959). Genes that were linked to these biological process terms included the major histocompatibility complex (MHC) class II members (*Cd74*, *RT1-Ba*, *RT1-Bb*, *RT1-Da*, *RT1-Db1*, and *RT1-Db2*) and their expression transactivator (*Ciita*), adhesion receptor (*Adgre1*), chemoattractants for B lymphocyte (*Cxcl13*), complement C3 (*C3*), neutrophil cytosolic factor 1 (*Ncf1*), proinflammatory cytokines (*Il1b*, *Tnfsf13b*), regulators of cellular responses to cytokines (*Csf2rb*), inflammatory response to IFNG/IFN-gamma (*Mefv*), catalyzer of the first step in leukotriene biosynthesis (*Alox5*), Ras-related C3 botulinum toxin substrate 2 (*Rac2*), and ubiquitin-like protein modifier (*Ubd*).

Contrary to our expectation, pretreatment with DEX did not significantly affect the responses of most of these immune/inflammatory-related genes to LPS. Figure 6 shows similar changes in the expression of several representative genes, such as MHCII genes and *Ciita* (a), *Il1b*, and *Tnfsf13b* (b) after LPS alone and with DEX pretreatment. The RNA-sec results were confirmed using quantitative PCR for the *IL1b* chosen for verification (c). A significant correlation (r = +0.93) was found between sec and PCR *IL1b* expression values.

Due to this, GO analysis (DAVID) of DEGs that were revealed between DEX + LPS and DEX + SAL also showed that the top eight biological processes (FDR < 0.05) were mainly related to immune response (Appendix A): “antigen processing and presentation of exogenous peptide antigen via MHC class II” (GO:0019886), “immunoglobulin production involved in immunoglobulin mediated immune response” (GO:0002381), “peptide antigen assembly with MHC class II protein complex” (GO:0002503), “antigen processing and presentation of peptide or polysaccharide antigen via MHC class II” (GO:0002504), “positive regulation of T cell activation” (GO:0050870), “antigen processing and presentation” (GO:0019882), “response to interferon-gamma” (GO:0034341), and “adaptive immune response” (GO:0002250).

To explore the candidate genes and pathways that might play important roles in preventing LPS-induced memory impairment by DEX, STRING (the Search Tool for the Retrieval of Interacting Genes/Proteins, https://string-db.org/) analyses were performed for common and unique DEGs between LPS vs. SAL and DEX + LPS vs. DEX + SAL.

Among the 33 DEGs from LPS vs. SAL, 3 genes (*Cd4*, *Cd74*, and *Il1b*) ranked with the interaction degree ≥ 10 connections/interactions, which served as the top hub genes (Table). Among DEGs from a comparison of DEX + LPS and DEX + SAL, the number of top hub genes was six, the list of which was similar to LPS vs. SAL-related DEGs (*Cd4*, *Cd74*, *Il1b*) and included *Ciita*, *Creb1*, and *RT1-Bb* (Table 1). A high level of connections (*n* = 9) was also observed for the *Grin2a* gene. Of these hub genes, *Cd4* exhibited the highest degree.

Of the 33 DEGs revealed between LPS and SAL, 9 did not fall into the protein–protein interaction (PPI) network (the number of connections/interactions = 0), which, when made from only other DEGs, was constructed of 24 nodes and 62 edges. K-means clustering of these DEGs outlined two clusters, as depicted in Figure 7a, and the biological functions of each were characterized by the Gene Ontology (GO). The cluster 1 (green) included 17 DEGs together with *Cd4* and *Cd74* hub genes (*Cd22*, *Cd4*, *Cd74*, *Ciita*, *Csf2rb*, *Cxcl13*, *Emr1*, *Itgal*, *Mx1*, *Plac8*, *RT1-Ba*, *RT1-Bb*, *RT1-Da*, *RT1-Db1*, *RT1-Db2*, *Rac2*, and *Tnfsf13b*). These DEGs were mainly enriched in immune response and immune system processes. The second cluster consisted of seven DEGs together with the *Il1b* hub gene (*Alox5*, *C3*, *F10*, *Ifnlr1*, *Il1b*, *Mefv*, and *Ncf1*), which were involved in the inflammatory response and response to stress.

From 22 DEGs shared between LPS vs. SAL and DEX + LPS vs. DEX + SAL, 16 DEGs, together with *Cd4*, *Cd74*, and *Il1b* hub genes (*Alox5*, *C3*, *Cd22*, *Cd4*, *Cd74*, *Ciita*, *Cxcl13*, *Emr1*, *Il1b*, *Itgal*, *Ncf1*, *RT1-Ba*, *RT1-Bb*, *RT1-Da*, *RT1-Db1*, and *Tnfsf13b*) that fell into the PPI network, were also predominantly related to immune and inflammatory responses (Figure 7b).

In addition to the similarity with LPS vs. SAL GO biological processes associated with immune-inflammatory activation that was revealed by DAVID analyses, DEGs from DEX + LPS vs. DEX + SAL were also enriched (*p*-value < 0.05) in the “regulation of ion transmembrane transport” (GO:0034765), “cellular response to growth factor stimulus” (GO:0071363), “regulation of membrane potential” (GO:0042391), “regulation of long-term neuronal synaptic plasticity” (GO:0048169), “potassium ion transmembrane transport” (GO:0071805), “intracellular signal transduction” (GO:0035556), and “calcium ion transport” (GO:0006816) (Appendix A). Genes associating with these terms changed their expression specifically in the LPS-administered rats pretreated with DEX, but not in rats that received LPS or DEX alone. The list of these genes included calcium channels (*Cacna1e* and *Cacna2d1*), cyclic AMP-responsive element-binding protein 1 (*Creb1*), gamma-aminobutyric acid receptor subunits (*Gabrb2* and *Gabrr2*), glutamate ionotropic receptor nmda type subunit 2a (*Grin2a*), metabotropic glutamate receptor 5 (*Grm5*), potassium voltage-gated channels (*Kcnh5*, *Kcnh7*, and *Kcnq3*), and mitogen-activated protein kinase kinase kinase 2 (*Map3k2*). Figure 8 illustrates the specific increasing effects of DEX on representative genes in LPS-treated rats.

Out of 84 specific DEGs for DEX + LPS vs. DEX + SAL DEGs, 39 DEGs were in the string PPI network. Using k-means clustering, 21 DEGs, including the *Grin2a* hub gene, were grouped (Figure 9). These genes were associated with GO biological processes, such as “Regulation of transmembrane transport” (GO:0034762), “Cognition” (GO:0050890), “Learning” (GO:0007612), “Neurogenesis” (GO:0022008), and “Nervous system development” (GO:0007399).

## 4. Discussion

The main finding of the present study is that pretreatment with DEX can affect processes associated with glutamatergic signaling and nervous system development, possibly involved in the recovery of memory impairment induced by LPS.

### 4.1. Modeling of Inflammation

LPS, a bacterial mimetic, is widely used as an effective approach to induce neuroinflammation [23] and sickness behavior in rodents [24,25]. Neuroinflammation resulting from systemic or intra-brain LPS administration can induce and aggravate central degeneration [16,26]. We infused LPS into the striatum, which is among the most damaged areas of the brain after stroke, as found in rat models of clinical relevance [27]. Striatum damage is often accompanied by secondary injury of the hippocampus [28]. Twenty-four hours after LPS administration, we found pro-inflammatory and pro-apoptotic activation in the part of the striatum that was the site of LPS injection, but not in the other part, supporting the previous results [15], in which the authors even suggested using the contralateral region as a control. However, in contrast to the striatum, our data demonstrated that short-term pro-inflammatory responses to intra-striatal LPS were similar in both the ipsilateral and contralateral hippocampi.

The activation of brain glial cells is considered a critical event for mediating the neuroinflammatory and neurodegenerative responses to LPS. In a recent study, 2-[18F]fluoro-2-deoxy-D-glucose (6-[18F]FDF) was used to specifically image microglia in the rat model of neuroinflammation [29]. During 2 days post-surgery (local injection of LPS into the right striatum), an increased uptake of 6-[18F]FDF in the injected side vs. the left striatum was found. However, in these experiments, the left and right hippocampus did not differ in 6-[18F]FDF uptake after LPS injection. In the injected striatal side and both hippocampi, we found a significant elevation in gene expression of the main pro-inflammatory cytokine *Il1b* at 24 h after endotoxin administration. In the hippocampus, this effect was associated with an increase in the expression of Iba-1 protein, a well-known marker for the activated microglia [30]. In the right striatum at 24 h after LPS, a significant increase in the number of microglial cells immunopositive for both Iba-1 and active caspase-3 was also found. An increase in the expression of caspase-3 may reflect its involvement in regulating microglia activation by endotoxin as, for example, was suggested from observations at 24 h after injection of LPS into the rat substantia nigra [31].

### 4.2. Acute and Delayed Behavioral Effects of LPS: Influence of DEX Pretreatment

At 24 h, the Garcia score (evaluating neurological deficit) of LPS-treated rats was significantly lower compared to the animals injected with SAL. It should be noted that the Garcia test was designed for assessing functions in rodents after ischemic brain injury [17]; however, this test is also used in other animal models, in the first place, in traumatic brain injury models [32,33]. A decrease in neurologic scores after brain injury can depend on LPS as they were markedly worse by the endotoxin [33]. In our study, pretreatment with such anti-inflammatory synthetic glucocorticoid as DEX attenuated an LPS-induced decrease in the neurological scores.

Brain damage resulting from injury or inflammatory disease can lead to the development of dementia, the most common symptom of which is memory loss [34]. However, in a short time after a single LPS systemic or intracerebroventricular injection, the data of acute behavioral effects are mixed indicating a decrease [9,35] or no changes [36] in the NOR test. One reason for these inconsistencies might be that specific neural processes underlying the impairment of this form of memory were not yet developed. In agreement with such possibility, rats spent significantly less time exploring the novel object than the control animals at day 70, but not at day 35 after the administration of LPS on postnatal days 7 and 9 [37]. The rats that were given intraperitoneal injections of LPS once a week for 7 weeks exhibited a delay-dependent impairment in spatial memory [38]. Moreover, even in the absence of ongoing neuroimmune activation, memory deficits including object recognition impairment can persist for at least eight weeks after the LPS intraperitoneal injection [39]. Similar to our preliminary study [10], in the present work, the intra-striatal LPS administration also resulted in object recognition task impairment at 3 months.

Given the accepted participation of inflammatory response in the neurocognitive disturbances and anti-inflammatory action of DEX, we thought that DEX could reduce delayed memory impairment after LPS. The results of the present study support this idea. In 3 months, the recognition index significantly decreased in the LPS group, and this effect was significantly attenuated by DEX pretreatment. This outcome is in agreement with data from an acute research. The impairment in the cognitive ability of mice in the Morris water maze test and development of neuroinflammation 24 h after ischemia-reperfusion were prevented by DEX when intraperitoneally administered 30 min before the ischemia [12].

### 4.3. Influence of DEX Pretreatment on Transcriptomic LPS Effects

For the first time, it was shown that a single intra-brain LPS injection upregulated and downregulated hippocampal gene expression 3 months later. DEGs after LPS administration were predominantly associated with immune and inflammatory responses. Some long-term gene and protein expression changes associated with peripheral LPS administration have been reported previously. Continued immune/inflammatory activation after LPS was in agreement with those from previous studies, in which even a single peripheral LPS injection can impact the central nervous system for a long time. For example, *TNFα* protein remained elevated in rodent brains [40], including the hippocampus [41], 10 months after LPS intraperitoneal administration. In another study, three months after the end of intraperitoneal LPS injection, several neuroimmune-related genes including *Cd74* also showed persistent upregulation in the hippocampus of male rats [42]. Although how LPS mediates memory impairment is still poorly understood, the changes in the expression of some inflammatory-related genes, especially *Il1b*, but not obviously in the expression of MHC II-related genes [43], may be theoretically linked with memory impairment. The novel object recognition performance in the NOR test was inversely correlated with the levels of IL1b in the hippocampus of rats one week after the intraperitoneal injection of LPS [44]. On the other hand, *IL1b* knock-down in the hippocampus significantly attenuated the memory deficits induced in mice by LPS injection [45]. However, in our study, the pretreatment of LPS-exposed rats with DEX, which led to an increase in recognition index, did not affect the expression of the IL1b gene suggesting that DEX involved some other mechanisms in the recovery of memory under neuroinflammatory conditions.

GO biological process enrichment analysis showed that *Cacna1e*, *Creb1*, *Grin2a*, and *Grm5* were significantly enriched in cognition and learning in our study. Changes in the expression of modulators of ion channels, particularly Ca^2+^ ion signaling, could be especially important. The administration of felodipine, the L-type Ca^2+^ channel blocker, to C57BL/6 mice for 9 days significantly reduced LPS-induced spatial memory impairment in the Y-maze test by modulating the formation of hippocampal dendritic spine, but it did not significantly change the novel object preference in the NOR test on days 8–9 of the treatment regimen [46].

The activating effects of glutamate, the main excitatory neurotransmitter in the mammalian brain, on post-synaptic neurons occur via ionotropic and metabotropic receptors and often include Ca^2+^ influx [47]. The functional properties of the Ionotropic receptors were determined by the composition of the subunits, each of which was encoded by a separate gene [48,49]. GRIN2A (Glutamate Ionotropic Receptor NMDA Type Subunit 2A)-containing hippocampal NMDARs (N-methyl-d-aspartate receptors) were shown to play an important role in synaptic plasticity, learning and memory [50]. These receptors were involved in object recognition memory reconsolidation [51], and GRIN2A subunit knockout mice exhibited discrimination learning impairments [52]. In light of these data, our findings of increased Grin2a expression along with improved memory in DEX-pretreated animals suggest the involvement of this receptor subunit in the prevention of endotoxin-induced long-term memory loss by glucocorticoids. The possible involvement of an increase in the expression of *Grm5* gene, coding metabotropic glutamate receptor 5, in the mechanisms of prevention of LPS-induced long-term memory decline by DEX is supported by numerous data suggesting the use of this receptor as a therapeutic target for improving memory deficits [53]. For example, aged rats exposed to one month of environmental enrichment demonstrated an increase in learning and memory in the Morris water maze and novel object recognition behavioral tasks that were accompanied by enhanced hippocampal function, including an increase in the level of mGluR5 [54]. In agreement with these data and reductions in hippocampal mGluR5 binding in early Alzheimer’s disease that were revealed with the help of positron emission tomography [55], our results may expand the understanding of Grm5’s role in memory disturbances and correction.

In addition to the receptors, in a study by Brothers with co-authors [56], the capacity of the brain to compensate for the presence of chronic neuroinflammation through enhanced clearance of extracellular glutamate was demonstrated. Excitotoxicity associated with glutamate accumulation in the extracellular space is regarded among the reasons for memory impairment. The increased expression of the *Slc1a2* gene encoding transporter for glutamate reuptake from the synaptic cleft was previously observed in the hippocampus one day after the single central administration of LPS [49], but not at 3 months in the present study. At the same time, in LPS-exposed rats pretreated with DEX, the expression in *Slc1a2* significantly increased 3 months after the drug administration, indicating enhanced glutamate clearance in DEX-pretreated animals.

The natural aging process is associated with an increased inflammatory response of the brain, so it is not surprising that this model is often used to analyze the mechanisms of memory impairment associated with inflammatory activation. Inflammation and cognitive decline were investigated in old rats and young animals repeatedly injected (once a week for 7 weeks) with LPS. Animals from both groups demonstrated similar increases in the expression of genes associated with neuroinflammation, as well as impaired spatial memory [38]. In this experiment, however, in contrast to the age-related decrease in the transcription of synaptic genes, the animals exhibited increased expression of genes after LPS that support the growth and maintenance of synapses, suggesting the involvement of some synaptic processes in response to inflammation. In our study, DEX pretreatment affected the expression of genes that were associated with biological processes such as neurogenesis (9 DEGs: *Cdkl5*, *Chl1*, *Creb1*, *Erbb4*, *Gabrb2*, *Grin2a*, *Grm5*, *Kcnq3*, and *Unc5d*) and nervous system development (10 DEGs: *Chl1*, *Cdkl5*, *Creb1*, *Erbb4*, *Gabrb2*, *Grin2a*, *Grm5*, *Kcnq3*, *Slc1a2*, and *Unc5d*). Among the proteins coded by listed genes, cyclin-dependent kinase-like 5 (CDKL5) was suggested as an important regulator of synaptic function in glutamatergic neurons, and the ablation of its expression impaired the hippocampal-dependent memory in mice [57]. The dysfunction of NRG1/ErbB4 signaling in the hippocampus was shown to mediate a long-term memory decline in a mouse model of systemic inflammation induced by repeated LPS injections [58].

In contrast to DEX transcriptomic effects in the LPS-treated group, no differences in the gene expression were observed between DEX + SAL and SAL groups. The poor penetration of dexamethasone into the brain [59] may be among the possible reasons for the lack of transcriptome response to glucocorticoid in the SAL group, whereas the well-known LPS-induced increase in the blood-brain barrier permeability appears to enhance brain susceptibility to DEX.

It should be also noted that the strong criteria for the screening of DEGs used in our study could lead to the loss of some functionally important genes, including, for example, the gene coding the receptor for glucocorticoids (GR). The comparison of the RNA-sec data for DEX + LPS and DEX + SAL showed a moderate increase in the expression of *Nr3c1* (log2(FC) = 0.24; *p*-value = 2.29 × 10^−4^; padj = 5.03 × 10^−3^) in DEX + LPS. Changes in the expression of GR that were involved in the regulation of the transcription of numerous genes may also play a role in the induction of DEX pretreatment effects, including memory recovery. For example, the protein encoded by *Ncf1* (Neutrophil Cytosolic Factor 1) is a cytosolic subunit of NADPH oxidase, which influences learning and memory via reactive oxygen species produced by hippocampal microglia [60]. NADPH oxidase activities, in turn, can be affected by GR ligands [61].

Overall, the results of the present study showed that pretreatment with DEX did not markedly affect LPS-induced prolonged inflammatory response, and the attenuating effect of glucocorticoid on cognitive impairment may occur partly through changes in the glutamatergic system.

## 5. Conclusions

Taken together, the data suggest that (1) pretreatment with DEX did not markedly affect LPS-induced prolonged inflammatory response; (2) DEX pretreatment can activate processes associated with glutamatergic signaling and nervous system development that may be involved in the recovery of the memory impairment induced by LPS.

## Figures and Tables

**Figure 1 biomedicines-11-02595-f001:**
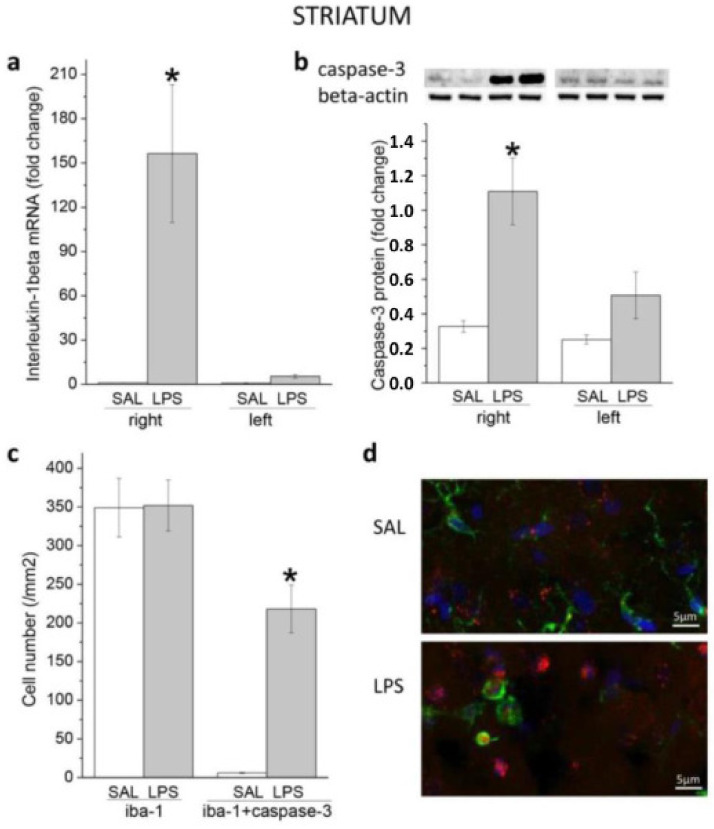
Pro-inflammatory and pro-apoptotic effects of LPS in the striatum. In the right striatum, which was the site of endotoxin infusion, LPS significantly (* *p* < 0.05) increased gene expression of pro-inflammatory cytokine interleukin-1beta (**a**) and protein of active caspase-3 (**b**) compared with SAL group at 24 h. In the right striatum, immunohistochemical analysis, using Iba-1 as a specific marker for microglia, did not reveal any differences between the LPS and SAL groups in the number of cells that were only Iba1 immuno-positive, whereas the number of cells immunoreactivities for both Iba-1 and caspase-3 was significantly increased (* *p* < 0.05) (**c**). (**d**) Iba-1 (green) and caspase-3 (red) immunofluorescence of representative images from the right striatum after SAL (up) and LPS (down).

**Figure 2 biomedicines-11-02595-f002:**
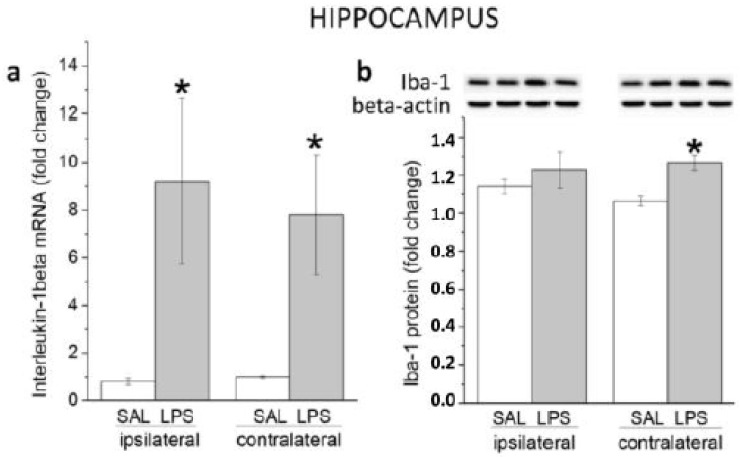
Expression levels of interleukin-1beta mRNA (**a**) and protein of Iba-1 (**b**) in ipsilateral and contralateral hippocampi 24 h following administration of LPS into right striatum. * *p* < 0.05 compared with an appropriate SAL group.

**Figure 3 biomedicines-11-02595-f003:**
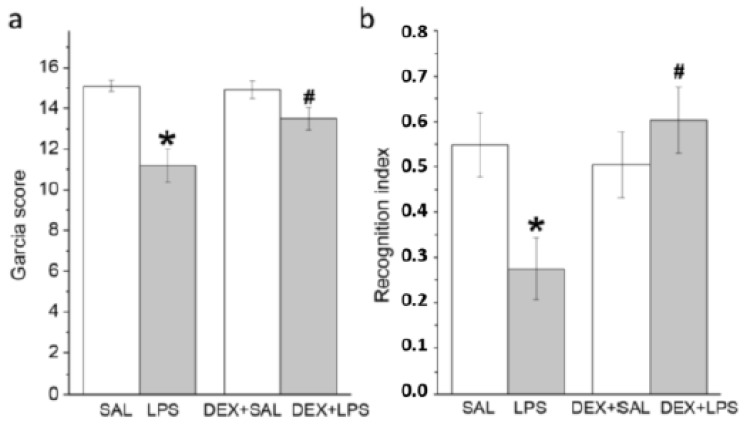
Short-term and delayed behavioral effects of LPS administered alone or with DEX. (**a**) The Garcia score at 24 h. Neurological scale of the LPS-treated rats significantly decreased compared to that of the SAL-treated rats. When animals were pretreated with DEX, the neurological scale of the DEX + LPS group was higher than after LPS alone. (**b**) The recognition index was assessed 3 months after the drug administration. LPS caused memory impairment, and this effect was prevented by DEX pretreatment. Each group of animals that were the same on (**a**,**b**) consisted of 10 rats. * *p* < 0.05 compared with an appropriate control group (SAL or DEX + SAL); # *p* < 0.05 compared with LPS group.

**Figure 4 biomedicines-11-02595-f004:**
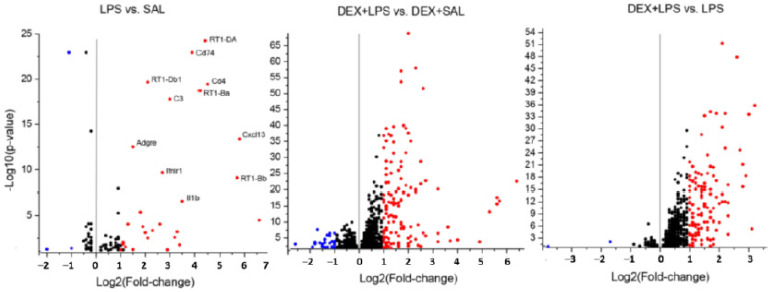
Volcano plots show the differentially expressed genes (padj < 0.05 for all dots). Red represents upregulated genes, and blue represents downregulated genes with │log2 (fold change)│ ≥ 1.

**Figure 5 biomedicines-11-02595-f005:**
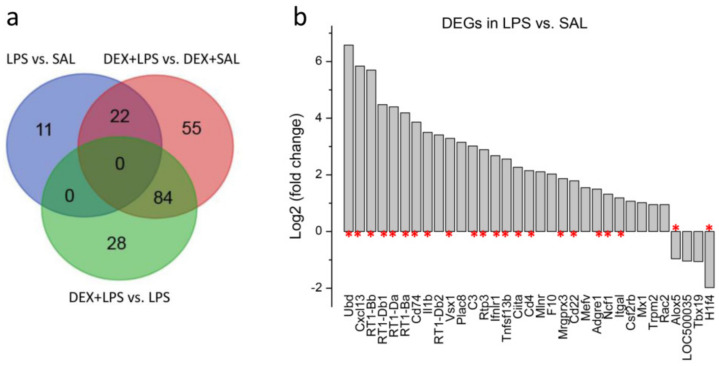
(**a**) Common genes among DEGs in LPS vs. SAL and DEX + LPS vs. DEX + SAL (22 genes), as well as DEX + LPS vs. DEX + SAL and DEX + LPS vs. LPS (84 genes). (**b**) Expression profiles of DEGs for LPS vs. SAL. Data were expressed as log2 transformation of fold changes. Asterisks indicate 22 DEGs common for LPS vs. SAL and DEX + LPS vs. DEX + SAL.

**Figure 6 biomedicines-11-02595-f006:**
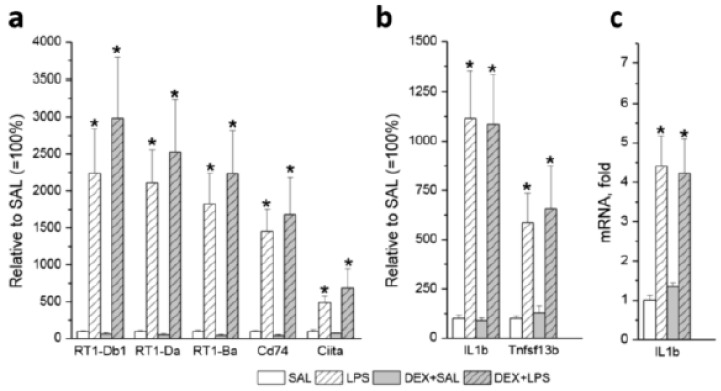
Responses of representative genes to LPS alone and with DEX pretreatment (relative to SAL group): (**a**) MHCII-related genes (*RT1-Db1*, *RT1-Da*, *RT1-Ba*, and *Cd74*), *Ciita*; (**b**) *Il1b* and *Tnfsf13b*; (**c**) verification of the RNA-sec results for *IL1b* by quantitative PCR. * *p* < 0.05 compared with an appropriate control group (SAL or DEX + SAL).

**Figure 7 biomedicines-11-02595-f007:**
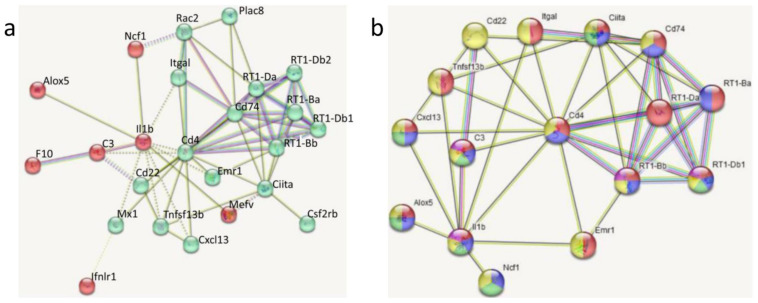
Protein–protein interaction (PPI) networks: (**a**) two k-means clusters of 24 DEGs between LPS and SAL; (**b**) network of the 16 DEGs shared between LPS vs. SAL and DEX + LPS vs. DEX + SAL. Colors indicate the association of DEGs with several GO biological processes: red—GO:0006955 immune response, 14 DEGs, FDR = 1.74 × 10^−11^; blue—GO:0006952 defense response, 11 DEGs, FDR = 1.63 × 10^−7^; green—GO:0006954 inflammatory response, 7 DEGs, FDR = 2.53 × 10^−5^; yellow—GO:0051716 cellular response to stimulus, 14 DEGs, FDR = 0.0024; violet—GO:0051384 response to glucocorticoid, 4 DEGs, FDR = 0.0048. FDR: false discovery rate.

**Figure 8 biomedicines-11-02595-f008:**
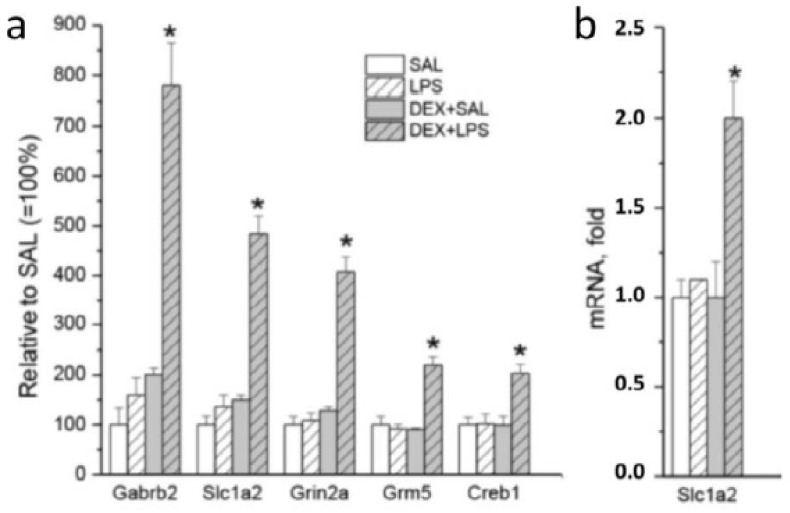
Responses of representative genes to LPS or DEX alone, as well as to LPS with DEX pretreatment (relative to SAL group). (**a**) Data from RNA-sec; (**b**) verification of the RNA-sec results for *Slc1a2* using quantitative PCR (correlation between sec and PCR expression values was 0.63, *p* < 0.05). * *p* < 0.05 compared with an appropriate control group (SAL or DEX + SAL).

**Figure 9 biomedicines-11-02595-f009:**
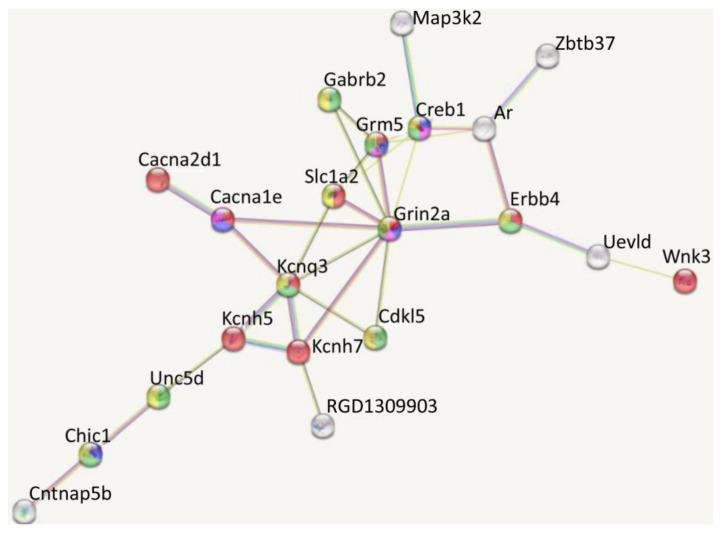
Protein–protein interaction (PPI) network of the DEGs specific for DEX + LPS vs. DEX + SAL (k-means cluster 1, 21 DEGs). Colors indicate the association of DEGs with several GO biological processes: red—GO:0034762 regulation of transmembrane transport, 10 DEGs (*Cacna1e*, *Cacna2d1*, *Erbb4*, *Grin2a*, *Grm5*, *Kcnh5*, *Kcnh7*, *Kcnq3*, *Slc1a2*, and *Wnk3*), FDR = 1.11 × 10^−6^; blue—GO:0050890 cognition, 5 DEGs (*Cacna1e*, *Chl1*, *Creb1*, *Grin2a*, and *Grm5*), FDR = 0.0165; violet—GO:0007612 learning, 4 DEGs (*Cacna1e*, *Creb1*, *Grin2a*, and *Grm5*), FDR = 0.0198; green—GO:0022008 neurogenesis, 9 DEGs (*Cdkl5*, *Chl1*, *Creb1*, *Erbb4*, *Gabrb2*, *Grin2a*, *Grm5*, *Kcnq3*, and *Unc5d*), FDR = 0.0223; yellow—GO:0007399 nervous system development, 10 DEGs (*Chl1*, *Cdkl5*, *Creb1*, *Erbb4*, *Gabrb2*, *Grin2a*, *Grm5*, *Kcnq3*, *Slc1a2*, and *Unc5d*), FDR = 0.0340. FDR: false discovery rate.

**Table 1 biomedicines-11-02595-t001:** The top genes ranked with the interaction degree method.

Gene	LPS vs. SAL (33)	DEX + LPS vs. DEX + SAL (161)	Common DEGs (22)	Specific DEGs (84)
*Cd4*	16	21	13	-
*Cd74*	11	14	8	-
*Ciita*	8	11	6	-
*Creb1*	-	11	-	7
*Il1b*	10	15	8	-
*RT1-Bb*	8	11	7	-
*RT1-Da*	9	9	6	-
*Grin2a*	-	9	-	9

## Data Availability

Data are available from the corresponding author upon request.

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
