# Peer review of "Genes Involved by Dexamethasone in Prevention of Long-Term Memory Impairment Caused by Lipopolysaccharide-Induced Neuroinflammation"

_biomedicines, 2023, doi:10.3390/biomedicines11102595_

Round 1
Reviewer 1 Report
Authors investigated short- and long-term effects of LPS induced neuroinflammation. Experiment is very interesting and well design and supported by adequate methodology. There are some comments Authors should consider.
title: 'Genes involved by dexamethasone..' sounds rather strange. Can Authors choose another term?
M&M: Did the rats (i.e. each rat) received total volume of 4ul of LPS in saline? It is not calculated per g of body weight?
Design of the research (short and long term effects) is very interesting and well thought out. It would be more convenient if Authors rearrange M&M sections..for example after describing design of the first part of the experiment (short term effect of LSP) Authors could describe used methods (western blot, mRNA, Garcia test..) and after describing second part of the experiment (long term effects) they could indicate appropriate methods.
Results: Authors should consider adding all representative IHC images to the supplementary material (we can see only right striatum where Authors report changes).
WB original images- Authors should indicate lanes (group) in the original images. They show representative wb images with 2 lanes/group but there are more lanes in the original images.
Discussion: 4.1. and 4.2. should include more discussion, not just repeating the results.
English Language used in the manuscript is adequate.
Author Response
Responses to Reviewer 1 Authors investigated short- and long-term effects of LPS induced neuroinflammation. Experiment is very interesting and well design and supported by adequate methodology. There are some comments Authors should consider. Comment: title: 'Genes involved by dexamethasone..' sounds rather strange. Can Authors choose another term? Response: We went through many options for the title of the manuscript until we settled on this one. It seems to us that it best reflects the essence of the article. Comment: M&M: Did the rats (i.e. each rat) received total volume of 4ul of LPS in saline? It is not calculated per g of body weight? Response: The dose of a biologically active substance is calculated per gram of body weight when it is administered peripherally, as in our case with dexamethasone, since it is assumed to be distributed throughout the body. Central infusion of the drug is aimed at inducing its local action, so the amount of drug administered is usually the same for all subjects. Comment: Design of the research (short and long term effects) is very interesting and well thought out. It would be more convenient if Authors rearrange M&M sections..for example after describing design of the first part of the experiment (short term effect of LSP) Authors could describe used methods (western blot, mRNA, Garcia test..) and after describing second part of the experiment (long term effects) they could indicate appropriate methods. Response: To avoid repeating the descriptions of some of the methods, such as PCR, used in short-term and long-term experiments, we find it convenient to use the original sections of the M&M. Comment: Results: Authors should consider adding all representative IHC images to the supplementary material (we can see only right striatum where Authors report changes). Response: We added representative IHC images of the right and left striatum to the supplementary material (Supplementary Materials: Additional Materials 2). They show that in contrast to the right striatum, no cells expressing active caspase-3 were found in the left striatum. Comment: Results: WB original images- Authors should indicate lanes (group) in the original images. They show representative wb images with 2 lanes/group but there are more lanes in the original images. Response: Yes, of course. Caspase-3 activation occurs in two-steps: zymogen is first cleaved by upstream caspases, and form active, p19/p12 complex; then autocatalytic process generates the fully mature p17/p12 form of the enzyme. The later form leads to death of cells while cytoplasmic active caspase-3 P19 is a trait of pro-inflammatory activated microglia (Kavanagh et al., 2014. Regulation of caspase-3 processing by cIAP2 controls the switch between pro-inflammatory activation and cell death in microglia. Cell Death Dis., 5:e1565, doi: 10.1038/cddis.2014.514). This pro-inflammatory-active caspase-3 P19 presented on immunoblot on the Figure 1 (b), and both forms – on the original blots. We added the full original images of western blots with legends to Supplementary Materials: Additional Materials 1. Comment: Discussion: 4.1. and 4.2. should include more discussion, not just repeating the results. Response: In 4.1., we added discussion of the data that in contrast to the striatum, short-term pro-inflammatory responses to endotoxin were similar in both the right and left hippocampus: “In recent study, 2-[18F]fluoro-2-deoxy-D-glucose (6-[18F]FDF) was used to specifically image microglia in the rat model of neuroinflammation [29]. During 2 days post-surgery (local injection of LPS into the right striatum), an increased uptake of 6-[18F]FDF in the injected side vs. the left striatum was found. However, in these experiments, the left and right hippocampus did not differ in 6-[18F]FDF uptake after LPS injection. Comments on the Quality of English Language English Language used in the manuscript is adequate.Reviewer 2 Report
At the manuscript “Genes involved by dexamethasone in prevention of long-term memory impairment caused by lipopolysaccharide-induced neuroinflammation” by Drs. Galina T. Shishkina et al. authors compared responses of hippocampal transcriptomes 3 months after the striatal infusion of lipopolysaccharide resulting in memory loss, or with dexamethasone pretreatment, which abolished the long-term LPS-induced memory impairment. The authors concluded that pretreatment with DEX did not markedly affect LPS-induced prolonged inflammatory response. However, DEX pretreatment can affect processes associated with glutamatergic signaling and nervous system development, possibly involved by that in the recovery of memory impairment induced by LPS
The authors did a lot of research work and obtained interesting data. The technique and the experimental paradigm do not cause objections, as well as the substantiation of the conclusions. I have only some questions.
The key point of the study is the release of microglia activation. There are quite a few publications on this topic, I would pay a little more attention to this phenomenon in the "discussion". Is there any reason to believe that this process occurs differently in the striatum than in the other parts of the brain? Some publications authors can cite:
Sumi et a;, Lipopolysaccharide-Activated Microglia Induce Dysfunction of the Blood–Brain Barrier in Rat Microvascular Endothelial Cells Co-Cultured with Microglia; Cell Mol Neurobiol. 2010; 30(2):247- 53. doi: 10.1007/s10571-009-9446-7
Kalyan M; Role of Endogenous Lipopolysaccharides in Neurological Disorders. Cells. 2022 Dec 14;11(24):4038. doi: 10.3390/cells11244038.
Authors reported, that in contrast to the striatum, short-term pro-inflammatory responses to endotoxin were similar in both the right and left hippocampus.
Is it related to the transcallosal connections? Or something else?
Minor criticisms
It would be nice to increase the size of the fonts in Fig 7A and Fig 9.
The presentation of a subject is systematic and comprehensive and analysis is proper. I am happy to recommend the manuscript for the publication after minor corrections mentioned above.
Author Response
Responses to Reviewer 2: Comments and Suggestions for Authors At the manuscript “Genes involved by dexamethasone in prevention of long-term memory impairment caused by lipopolysaccharide-induced neuroinflammation” by Drs. Galina T. Shishkina et al. authors compared responses of hippocampal transcriptomes 3 months after the striatal infusion of lipopolysaccharide resulting in memory loss, or with dexamethasone pretreatment, which abolished the long-term LPS-induced memory impairment. The authors concluded that pretreatment with DEX did not markedly affect LPS-induced prolonged inflammatory response. However, DEX pretreatment can affect processes associated with glutamatergic signaling and nervous system development, possibly involved by that in the recovery of memory impairment induced by LPS. The authors did a lot of research work and obtained interesting data. The technique and the experimental paradigm do not cause objections, as well as the substantiation of the conclusions. I have only some questions. Question: The key point of the study is the release of microglia activation. There are quite a few publications on this topic, I would pay a little more attention to this phenomenon in the "discussion". Is there any reason to believe that this process occurs differently in the striatum than in the other parts of the brain? Some publications authors can cite: Sumi et al, Lipopolysaccharide-Activated Microglia Induce Dysfunction of the Blood–Brain Barrier in Rat Microvascular Endothelial Cells Co-Cultured with Microglia; Cell Mol Neurobiol. 2010; 30(2):247- 53. doi: 10.1007/s10571-009-9446-7 Kalyan M; Role of Endogenous Lipopolysaccharides in Neurological Disorders. Cells. 2022 Dec 14;11(24):4038. doi: 10.3390/cells11244038. Response: Neuroinflammation resulted from microglial activation is assumed as a common in neurodegenerative diseases. This process theoretically can occur differently in the striatum than in the other parts of the brain. For example, region-dependent down- and up-regulation of selected mRNAs was observed in the putamen and frontal cortex in cases with Parkinson disease-related pathology (Garcia-Esparcia P et al. Complex deregulation and expression of cytokines and mediators of the immune response in Parkinson's disease brain is region dependent. Brain Pathol. 2014; 24(6):584-98). Such possibility is supported by data about differential regional effects in gene expression and activated pathways involved in inflammatory-related signaling under stressful conditions (Adkins AM et al. Stressor control and regional inflammatory responses in the brain: regulation by the basolateral amygdala. J Neuroinflammation. 2023; 20(1): 128). The origins of regional differences are largely unknown, and further research is needed to elucidate this complex issue. In part, they may be related to the region-specific homeostatic transcriptional identity exhibited by both microglia and astrocytes (Grabert K et al. Microglial brain region-dependent diversity and selective regional sensitivities to aging. Nat Neurosci. 2016;19(3):504-16; Makarava N et al. Region-specific glial homeostatic signature in prion diseases is replaced by a uniform neuroinflammation signature, common for brain regions and prion strains with different cell tropism. Neurobiol Dis. 2020;137: 104783). Question: Authors reported, that in contrast to the striatum, short-term pro-inflammatory responses to endotoxin were similar in both the right and left hippocampus. Is it related to the transcallosal connections? Or something else? Response: Numerous studies have reported the rapid pro-inflammatory responses to LPS in different brain regions, however little attention has been given to the point for possible regional dependence of their initiation (Furube E et al. Brain Region-dependent Heterogeneity and Dose-dependent Difference in Transient Microglia Population Increase during Lipopolysaccharide-induced Inflammation. Sci Rep. 2018;8(1):2203.). In our study, in contrast to the striatum, short-term pro-inflammatory responses to endotoxin were similar in both the right and left hippocampus, indicating the specificity of LPS-induced local response, the mechanisms of which, including the role of transcallosal connections, remain unknown. Recent study, using 2-[18F]fluoro-2-deoxy-D-glucose (6-[18F]FDF) to image microglia, revealed an increased 6-[18F]FDF uptake in the LPS-injected right striatum vs. the left side at 48 h post-surgery in both male and female rats, whereas hippocampus did not show any significant left and right side difference in this uptake (Boyle AJ et al. PET Imaging of Fructose Metabolism in a Rodent Model of Neuroinflammation with 6-[18F]fluoro-6-deoxy-D-fructose. Molecules. 2022; 27(23): 8529). We added the last data in the Discussion section 4.1.: “In recent study, 2-[18F]fluoro-2-deoxy-D-glucose (6-[18F]FDF) was used to specifically image microglia in the rat model of neuroinflammation [29]. During 2 days post-surgery (local injection of LPS into the right striatum), an increased uptake of 6-[18F]FDF in the injected side vs. the left striatum was found. However, in these experiments, the left and right hippocampus did not differ in 6-[18F]FDF uptake after LPS injection.” Minor criticisms: It would be nice to increase the size of the fonts in Fig 7A and Fig 9. Response: The size of the fonts in Fig 7A and Fig 9 was increased.